# *APOL1* renal risk variants have contrasting resistance and susceptibility associations with African trypanosomiasis

Anneli Cooper[1†], Hamidou Ilboudo[2,3†], V Pius Alibu[3,4], Sophie Ravel[5], John Enyaru[3,4], William Weir[1], Harry Noyes[1,3,6], Paul Capewell[1], Mamadou Camara[3,7], Jacqueline Milet[5], Vincent Jamonneau[2,3,5], Oumou Camara[7], Enock Matovu[3,8], Bruno Bucheton[3,5,7†], Annette MacLeod[1,3*†]

[1]Wellcome Trust Centre for Molecular Parasitology, College of Medical, Veterinary and Life Sciences, University of Glasgow, Glasgow, United Kingdom; [2]Centre International de Recherche-Développement sur l'Elevage en zone Subhumide, Bobo-Dioulasso, Burkina Faso; [3]TrypanoGEN, H3Africa Consortium, Makerere University, Kampala, Uganda; [4]College of Natural Sciences, Makerere University, Kampala, Uganda; [5]Unité Mixte de Recherche IRD-CIRAD 177, Institut de Recherche pour le Développement, Montpellier, France; [6]Institute of Integrative Biology, University of Liverpool, Liverpool, United Kingdom; [7]Programme National de Lutte contre la Trypanosomiase Humaine Africaine, Conakry, Guinea; [8]College of Veterinary Medicine, Animal Resources and Biosecurity, Makerere University, Kampala, Uganda

*For correspondence: annette. macleod@glasgow.ac.uk

†These authors contributed equally to this work

Competing interests: The authors declare that no competing interests exist.

**Abstract** Reduced susceptibility to infectious disease can increase the frequency of otherwise deleterious alleles. In populations of African ancestry, two *apolipoprotein-L1 (APOL1)* variants with a recessive kidney disease risk, named G1 and G2, occur at high frequency. APOL1 is a trypanolytic protein that confers innate resistance to most African trypanosomes, but not *Trypanosoma brucei rhodesiense* or *T.b. gambiense,* which cause human African trypanosomiasis. In this case-control study, we test the prevailing hypothesis that these *APOL1* variants reduce trypanosomiasis susceptibility, resulting in their positive selection in sub-Saharan Africa. We demonstrate a five-fold dominant protective association for G2 against *T.b. rhodesiense* infection. Furthermore, we report unpredicted strong opposing associations with *T.b. gambiense* disease outcome. G2 associates with faster progression of *T.b. gambiense* trypanosomiasis, while G1 associates with asymptomatic carriage and undetectable parasitemia. These results implicate both forms of human African trypanosomiasis in the selection and persistence of otherwise detrimental *APOL1* kidney disease variants.

## Introduction

Infectious disease is a major driving force of natural selection on human populations. Such evolutionary pressures can select for genetic variants that confer increased resistance to infectious agents, but may also predispose to specific genetic disorders, as exemplified by *Plasmodium* selection for the sickle-cell trait (*Allison, 1954*). Like sickle-cell disease, chronic kidney disease also affects millions worldwide (*Global Burden of Disease Study 2013 Collaborators, 2015*), with a disproportionate risk in populations of recent sub-Saharan African ancestry (*National Institutes of Health and National Institute of Diabetes and Digestive and Kidney Diseases, 2010*; *Norris and Agodoa,*

**eLife digest** African-Americans have a greater risk of developing chronic kidney disease than Americans with European ancestry. Much of this increased risk is explained by two versions of a gene called *APOL1* that are common in people with African ancestry. These two versions of the gene, known as G1 and G2, suddenly became much more common in people in sub-Saharan Africa in the last 10,000 years. One theory for their rapid spread is that they might protect against a deadly parasitic disease known as African sleeping sickness. This disease is caused by two related parasites of a species known as *Trypanosoma brucei*, one of which is found in East Africa, while the other affects West Africa.

Laboratory studies have shown that blood from individuals who carry the G1 and G2 variants is better at killing the East African parasites. However, it is not clear if these gene versions help people living in the rural communities, where African sleeping sickness is common, to fight off the disease.

Now, Cooper, Ilboudo et al. show that G1 and G2 do indeed influence how susceptible individuals in these communities are to African sleeping sickness. Individuals with the G2 version were five-times less likely to get the disease from the East African parasite. Neither version could protect individuals from infection with the West African parasite, but infected individuals with the G1 version had fewer parasites in their blood and were less likely to become severely ill. The ability of the G1 version to control the disease and prolong life could explain why this gene version has become so common amongst people in West Africa.

Unexpectedly, the experiments also revealed that people with the G2 version were more likely to become severely unwell when they were infected by the West African parasite. This indicates that whether this gene variant is helpful or harmful depends on where an individual lives. The next step following on from this work will be to investigate exactly how the G1 version reduces the severity of the West African disease. This may aid the development of new drugs for African sleeping sickness and kidney disease.

*2002*; *McClellan et al., 1988*). In African-Americans a large component of this disparity has been attributed to two common genetic variants of *APOL1* (MIM 603743), known as G1 and G2 (*Genovese et al., 2010*; *Tzur et al., 2010*). These variants are closely spaced in the C-terminal domain of *APOL1* but are located on separate haplotypes (*Genovese et al., 2010*) (*Figure 1*).

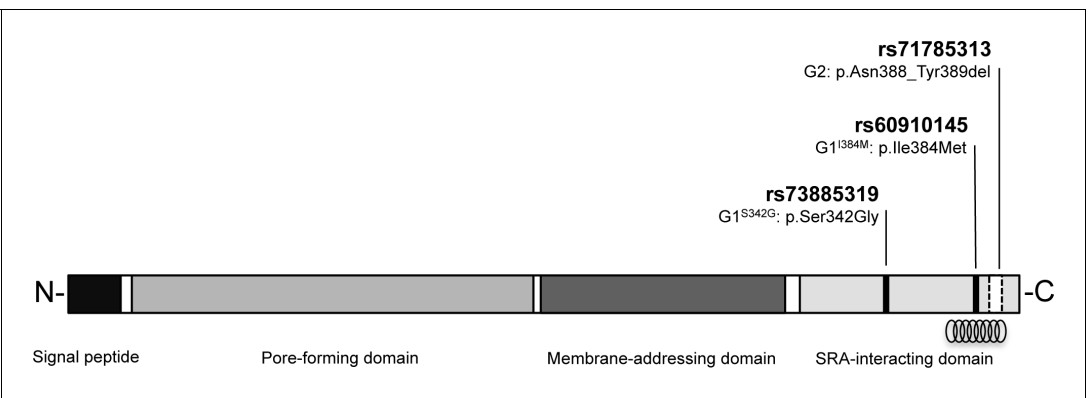

**Figure 1.** Schematic of G1 and G2 polymorphisms in human apolipoprotein L1. Human apolipoprotein-L1 (APOL1) is a 398-amino acid protein consisting of a cleavable N-terminal signal peptide, a pore-forming domain, a membrane-addressing domain, and a serum resistance-associated (SRA)-interacting domain. The polymorphisms that characterize the G1 and G2 renal risk variants are located in the SRA-interacting domain, the target site for binding of the SRA protein expressed by the human-infective *T.b.rhodesiense* parasite, which results in loss of APOL1 lytic function. The location of the critical binding region (residues 370–392) for this interaction is indicated by a helical graphic. G1 consists of two missense SNPs rs73885319 (p. Ser342Gly) and rs60910145 (p.Ile384Met) while the G2 polymorphism, rs71785313 (p.Asn388_Tyr389del), is found on an alternative *APOL1* haplotype, and represents an in-frame two amino acid deletion.

Individuals possessing a high-risk G1/G1, G2/G2 or G1/G2 genotype composed of two risk alleles (approximately 13% of African-Americans [*Friedman et al., 2011*]), are strongly predisposed to a wide spectrum of chronic kidney disorders that includes focal segmental glomerulosclerosis (*Genovese et al., 2010*; *Kopp et al., 2011*), HIV-associated nephropathy (*Kopp et al., 2011*; *Kasembeli et al., 2015*) and end-stage renal disease (*Genovese et al., 2010*; *Tzur et al., 2010*; *Freedman et al., 2014*). *APOL1* G1 and G2 are prevalent only in populations of recent African heritage (*Genovese et al., 2010*; *Kopp et al., 2011*), with evidence for a selective sweep within the last 10,000 years (*Genovese et al., 2010*), indicative of strong positive selection. Human African trypanosomiasis (HAT), a deadly parasitic disease endemic to sub-Saharan Africa, has been proposed as the source of this positive selective pressure (*Genovese et al., 2010*). HAT is caused by two tsetse fly-transmitted African trypanosomes, *Trypanosoma brucei rhodesiense* and *T.b. gambiense,* which are responsible for the acute East African form and more chronic West Africa form of the disease, respectively (*Kennedy, 2013*). Both parasites have been responsible for widespread fatal epidemics in sub-Saharan Africa throughout recorded human history (*Steverding, 2008*) suggesting the potential to exert potent selection pressure on the human genome. A heterozygous advantage model has been proposed for *APOL1* G1 and G2 (*Genovese et al., 2010*) in which recessive susceptibility to chronic kidney disease is balanced by dominant resistance to one or both forms of human African trypanosomiasis.

Prior to the discovery of its association with kidney disease, *APOL1* had already been recognised for encoding the pore-forming serum protein Apolipoprotein L1, which inserts into trypanosome membranes and effectively lyses the *Trypanosoma* species that cause disease in animals (*Vanhamme et al., 2003*; *Pérez-Morga et al., 2005*; *Molina-Portela et al., 2005*; *Thomson and Finkelstein, 2015*; *Vanwalleghem et al., 2015*). However, the two human-infective subspecies have evolved independent mechanisms to resist APOL1-mediated lysis. In *T.b. rhodesiense,* this is the result of an APOL1-binding protein (*Xong et al., 1998*; *De Greef et al., 1989*) whereas for *T.b. gambiense* the mechanism of APOL1 resistance appears more complex and multifactorial (*Capewell et al., 2013*; *Uzureau et al., 2013*; *DeJesus et al., 2013*; *Kieft et al., 2010*). It has been hypothesised that *APOL1* G1 and G2 variants could overcome one or more of these resistance mechanisms to protect against HAT. Indeed, previous studies have shown that *APOL1* G2 (and to a lesser extent G1) plasma is lytic to East African *T.b. rhodesiense* parasites *in vitro* (*Genovese et al., 2010*), but not West African *T.b. gambiense* (*Genovese et al., 2010*). Consequently, *T.b. rhodesiense* is considered the most likely candidate for positive selection of both *APOL1* variants in African populations (*Genovese et al., 2010*). Notably, however, the G1 variant appears significantly less effective at killing *T.b. rhodesiense*, and is found at very high frequency in West Africa (*Genovese et al., 2010*; *Kopp et al., 2011*; *Ko et al., 2013*; *Thomson et al., 2014*), where only *T.b. gambiense* is endemic (*Simarro et al., 2010*).

Furthermore, a class of asymptomatic individuals has been recently identified in *T.b. gambiense* disease foci, who exhibit a long-term *T.b. gambiense*-specific serological response but low or undetectable parasitemia indicative of a latent asymptomatic infection (*Koffi et al., 2006*; *Jamonneau et al., 2012*; *Bucheton et al., 2011*; *Ilboudo et al., 2011*; *Jamonneau et al., 2010*; *Garcia et al., 2000*). Parasites from such individuals appear genetically indistinguishable from those of *T.b. gambiense* clinical cases (*Kaboré et al., 2011*), suggesting disease outcome may be mediated by, as yet unidentified, host genetic factors. Field studies are therefore warranted to fully evaluate the contribution of variants of the host protein APOL1 to HAT susceptibility.

Here, we present a retrospective association study to test the relationship between *APOL1* G1 and G2 variants and susceptibility to the two different forms of human African trypanosomiasis, *T.b. rhodesiense* in East Africa and *T.b. gambiense* in West Africa. In Uganda, an association analysis was performed between *T.b. rhodesiense*-infected individuals and controls in a major disease focus. In the principal *T.b. gambiense* focus in Guinea, the presence of both clinical patients and asymptomatic individuals permitted a two-stage analysis. Firstly, the association between *APOL1* variants and susceptibility to *T.b. gambiense* infection (infected versus controls), and secondly the association with disease outcome following infection (clinical cases versus asymptomatic carriage). We report that the association of *APOL1* chronic kidney disease variants with HAT susceptibility are markedly different for the two subspecies. As hypothesised, a dominant protective association was detected for the G2 variant against *T.b. rhodesiense* infection. Conversely, we found that the *APOL1* G1 variant was not associated with resistance to *T.b. rhodesiense* infection, but with protective

asymptomatic carriage of *T.b. gambiense*. We consider the implications of these strikingly different susceptibilities in the context of human co-evolution with African trypanosomes and the distribution, selection and persistence of these kidney disease risk variants in sub-Saharan Africa.

## Results

### *APOL1* variants and resistance/susceptibility to *T.b. rhodesiense*

To test the heterozygous advantage hypothesis proposed for these *APOL1* variants against *T.b. rhodesiense* infection (*Genovese et al., 2010*), 180 controls and 184 clinically confirmed *T.b. rhodesiense* patients from a principle disease focus in central-eastern Uganda were genotyped for G1 and G2 polymorphisms. The G1 haplotype comprises of two non-synonymous substitutions, rs73885319 and rs60910145 situated just 128 bp apart and in near-perfect linkage disequilibrium (*Genovese et al., 2010*; *Kopp et al., 2011*). In this study, as reported by others (*Kopp et al., 2011*; *Behar et al., 2011*), a small number of individuals were identified with only a partial G1 haplotype (the kidney disease risk genotype at one of the G1 polymorphism positions but the non-risk genotype at the other) and were excluded from the G1 haplotype association analysis. The second chronic kidney disease risk variant, G2 (rs71785313), is found on an alternative haplotype and represents a six base pair in-frame deletion.

Comparing genotype frequencies in confirmed *T.b. rhodesiense*-infected individuals with uninfected controls found no association between the G1 haplotype and *T.b. rhodesiense* infection (p=0.50; *Table 1*). In contrast, we observed a significant dominant protective association for the G2 variant, with an odds ratio of 0.20 (95% CI: 0.07 to 0.48, p=0.0001; *Table 1*). This indicates a five-fold reduced susceptibility to *T.b. rhodesiense* infection for individuals that possess a single copy of the G2 variant, compatible with a model of heterozygous protection.

### *APOL1* variants and resistance/susceptibility to *T.b. gambiense*

#### Infection

To evaluate the impact of these polymorphisms in driving resistance/susceptibility to *T.b. gambiense* infection, G1 and G2 polymorphisms were genotyped in 227 *T.b. gambiense*-infected individuals and 104 controls from the mangrove focus in Guinea. When compared to control genotype frequencies, neither variant demonstrated an association with susceptibility to *T.b. gambiense* infection (p=0.47 [G1], p=0.50 [G2], *Table 2*).

#### Disease outcome

Infection with *T.b. gambiense* is associated with distinct clinical outcomes. *T.b. gambiense*-infected individuals can be subdivided into clinical stage trypanosomiasis patients who are serology and microscopy positive for trypanosomiasis (n = 167), and latent carriers (n = 60), who are defined as strongly serology positive, but asymptomatic and aparasitemic by microscopic examination for a period of at least two years. When comparing clinical cases and latent carriers, significant opposing associations were observed for the two *APOL1* variants with *T.b. gambiense* infection outcome (*Table 3*). There was an association between the G1 variant and dominant protection against developing clinical stage trypanosomiasis (OR = 0.33, 95% CI: 0.17–0.62; p=0.0005). This indicates that *T. b. gambiense*-infected individuals possessing a copy of the G1 *APOL1* variant were three-fold more likely to be latent asymptomatic carriers of *T.b. gambiense*. In contrast, the G2 variant was associated with a three-fold increased susceptibility to clinical stage trypanosomiasis (OR = 3.08, 95% CI: 1.45–7.06, p=0.0025), consistent with a risk of faster disease progression. This association was strengthened still further (OR = 5.87, 95% CI: 2.16–20.01; p=0.0001) when individuals with a potentially antagonistic compound heterozygous (G1/G2) genotype were excluded from the analysis (*Table 4*).

Together these associations, summarised in *Figure 2A*, indicate that the G1 and G2 *APOL1* variants exhibit distinct subspecies-specific susceptibility profiles in relation to the two causative agents of human African trypanosomiasis. The G1 variant is associated with asymptomatic carriage of *T.b. gambiense*, but the predicted protection against *T.b. rhodesiense* infection (*Genovese et al., 2010*) was not detected. For the G2 variant, opposing dominant associations were observed with the two

**Table 1.** Association between *APOL1* kidney disease risk variants and *T.b. rhodesiense* infection

| *APOL1* haplotype | Dominant model - Infection | | | | | |
|---|---|---|---|---|---|---|
| | *T.b.r* infected | | Control | | Association analysis*<br>*T.b.r* infected/Control | |
| | Number | % | Number | % | OR [95% CI] | P |
| G0 Ancestral Haplotype<br>rs73885319 (A) + rs60910145 (T) + rs71785313 (TTATAA) | | | | | | |
| G0 | 184 | 100.0 | 179 | 99.4 | N.C | 0.49 |
| Non-G0 | 0 | 0.0 | 1 | 0.6 | | |
| Total | 184 | 100.0 | 180 | 100.0 | | |
| G1 Haplotype[†]<br>rs73885319 (A>G) + rs60910145 (T>G) | | | | | | |
| G1 | 9 | 4.9 | 12 | 6.7 | 0.73 [0.29 to 1.79] | 0.50 |
| Non-G1 | 173 | 95.1 | 168 | 93.3 | | |
| Total | 182 | 100.0 | 180 | 100.0 | | |
| G2 Haplotype<br>rs71785313 (TTATAA>del6) | | | | | | |
| G2 | 6 | 3.3 | 26 | 14.4 | 0.20 [0.07 to 0.48] | 0.0001 |
| Non-G2 | 178 | 96.7 | 154 | 85.6 | | |
| Total | 184 | 100.0 | 180 | 100.0 | | |

*Two-tailed Fisher's exact test with mid-P method using a dominant genetic model (carriage of 1 or 2 copies of the designated *APOL1* haplotype),

[†]Individuals with only a partial G1 haplotype were excluded from the analysis. *T.b.r: T.b. rhodesiense*, OR: odds ratio, CI: confidence interval, N.C: not calculable. All raw data for **Table 1** can be found in **Table 1—source data 1**. The association analysis of the two individual component SNPs of the G1 haplotype can be found in **Table 1—source data 2**.

Source data 1. *APOL1* genotype data for *T.b. rhodesiense*-infected individuals and controls *Individuals excluded from the *APOL1* G1 association analysis. *T.b.r: T.b. rhodesiense*, G0: genotype compatible with the non-risk G0 allele for both rs73885319 and rs60910145, G1: genotype compatible with the G1 CKD risk allele for both rs73885319 and rs60910145, G1[M]: genotype compatible with the G1 CKD risk allele for rs60910145 and the non-risk G0 allele for rs73885319, G1[G]: genotype compatible with the G1 CKD risk allele for rs73885319 and the non-risk G0 allele for rs60910145, G2: genotype compatible with the G2 CKD risk allele for rs71785313.

Source data 2. Association between individual *APOL1* G1 kidney disease risk variants and *T.b. rhodesiense* infection Two-tailed Fisher's exact test with mid-P method using a dominant genetic model (carriage of 1 or 2 copies of the designated *APOL1* SNP). CKD: chronic kidney disease, *T.b.r: T.b. rhodesiense*, OR: odds ratio, CI: confidence interval. Raw data for **Source data 2** can be found in **Source data 1**.

different subspecies. This association is protective against *T.b. rhodesiense* infection, but with increased susceptibility to a more severe disease outcome for *T.b. gambiense*.

## Geographical distribution of *APOL1* G1 and G2 variants

To visualize the geographic distribution of *APOL1* variants in relation to HAT endemicity (*Figure 2B*), data generated by this study were merged with previously reported allele frequencies for 38 other sub-Saharan African populations, to produce a cohort of 5287 genotyped samples. Frequency distributions for G1 and G2 were transformed into geographical contour maps using the Kriging algorithm for data interpolation (*Figure 2C and D*). The allele frequencies from the Ugandan population and the mangrove foci in Guinea appear consistent with the general geographical distribution pattern for these variants in sub-Saharan Africa. Both variants are reported at higher prevalence in *T.b. gambiense* endemic West Africa, particularly G1, which reaches frequencies as high as 49% in the Ibo (*Thomson et al., 2014*) and Esan (*Abecasis et al., 2012*) tribes of Nigeria, decreasing to complete absence in Northeast Africa (*Tzur et al., 2010*; *Behar et al., 2011*). Allele frequency is moderately inversely correlated with longitude for both G1 (Pearson correlation: $r = -0.526$, $p = 2.0 \times 10^{-4}$, $N = 40$) and G2 ($r = -0.593$, $p = 5.7 \times 10^{-5}$, $N = 37$) but not latitude (G1, $p = 0.33$, G2,

**Table 2.** Association between kidney disease risk variants and *T.b. gambiense* infection

| APOL1 haplotype | Dominant model - Infection | | | | | |
| | *T.b.g* infected | | Control | | Association analysis[*] *T.b.g* infected/Control | |
| | Number | % | Number | % | OR [95% CI] | P |
|---|---|---|---|---|---|---|
| G0 Ancestral Haplotype rs73885319 (A) + rs60910145 (T) + rs71785313 (TTATAA) | | | | | | |
| G0 | 196 | 86.3 | 89 | 85.6 | 1.07 [0.54 to 2.06] | 0.84 |
| Non-G0 | 31 | 13.7 | 15 | 14.4 | | |
| Total | 227 | 100.0 | 104 | 100.0 | | |
| G1 Haplotype[†] rs73885319 (A>G) + rs60910145 (T>G) | | | | | | |
| G1 | 73 | 33.5 | 30 | 29.4 | 1.21 [0.73 to 2.03] | 0.47 |
| Non-G1 | 145 | 66.5 | 72 | 70.6 | | |
| Total | 218 | 100.0 | 102 | 100.0 | | |
| G2 Haplotype rs71785313 (TTATAA>del6) | | | | | | |
| G2 | 68 | 30.0 | 35 | 33.7 | 0.84 [0.51 to 1.40] | 0.50 |
| Non-G2 | 159 | 70.0 | 69 | 66.3 | | |
| Total | 227 | 100.0 | 104 | 100.0 | | |

[*]Two-tailed Fisher's exact test with mid-P method using a dominant genetic model (carriage of 1 or 2 copies of the designated *APOL1* haplotype),

[†]Individuals with a partial G1 haplotype were excluded from the analysis. *T.b.g: T.b. gambiense*, OR: odds ratio, CI: confidence interval. All raw data for **Table 2** can be found in **Table 2—source data 1**. The association analysis of the two individual component SNPs of the G1 haplotype can be found in **Table 2—source data 2**.

**Source data 1.** *APOL1* genotype data for *T.b. gambiense*-infected individuals and controls [*]Individuals excluded from the *APOL1* G1 association analysis. *T.b.g: T.b. gambiense*, G0: genotype compatible with the non-risk G0 allele for both rs73885319 and rs60910145, G1: genotype compatible with the G1 CKD risk allele for both rs73885319 and rs60910145, G1$^M$: genotype compatible with the G1 CKD risk allele for rs60910145 and the non-risk G0 allele for rs73885319, G1$^G$: genotype compatible with the G1 CKD risk allele for rs73885319 and the non-risk G0 allele for rs60910145, G2: genotype compatible with the G2 CKD risk allele for rs71785313.

**Source data 2.** Association between individual *APOL1* G1 kidney disease risk variants and *T.b. gambiense* infection Two-tailed Fisher's exact test with mid-P method using a dominant genetic model (carriage of 1 or 2 copies of the designated *APOL1* SNP). CKD: chronic kidney disease, *T.b.g: T.b. gambiense*, OR: odds ratio, CI: confidence interval. Raw data for **Source data 2** can be found in **Source data 1**.

p=0.30), indicating a significant decreasing relative frequency for both *APOL1* variants from West to East across the continent.

## Discussion

Here, we report the first association study between *APOL1* G1 and G2 kidney disease risk variants and *T.b. rhodesiense* and *T.b. gambiense*, revealing a more complex relationship with human African trypanosomiasis susceptibility than was originally predicted. The implications of these findings in relation to each of the human-infective parasite subspecies are considered in turn.

The zoonotic *T.b. rhodesiense* parasite has been responsible for a number of severe HAT outbreaks in recent human history in East Africa that have claimed hundreds of thousands of lives (**Hide, 1999**; **Fèvre et al., 2004**). Our data indicate that the *APOL1* G2 variant was strongly associated with protection against *T.b. rhodesiense* infection in a Ugandan disease focus. The observed five-fold reduced susceptibility for individuals possessing a single copy of the *APOL1* G2 variant is consistent with laboratory studies reporting in vitro lysis of *T.b. rhodesiense* for APOL1 G2 plasma and recombinant protein (**Genovese et al., 2010**), and increased survival of *APOL1* G2 transgenic mice in a *T.b. rhodesiense* infection model (**Thomson et al., 2014**). *T.b. rhodesiense* parasites are

**Table 3.** Association between kidney disease risk variants and *T.b. gambiense* infection outcome

| APOL1 haplotype | Dominant model – infection outcome | | | | | |
| | T.b.g Disease | | T.b.g Carriage | | Association analysis[*] T.b.g Disease/Carriage | |
| | Number | % | Number | % | OR [95% CI] | P |
|---|---|---|---|---|---|---|
| G0 Ancestral Haplotype rs73885319 (A) + rs60910145 (T) + rs71785313 (TTATAA) | | | | | | |
| G0 | 144 | 86.2 | 52 | 86.7 | 0.96 [0.38 to 2.25] | 0.95 |
| Non-G0 | 23 | 13.8 | 8 | 13.3 | | |
| Total | 167 | 100.0 | 60 | 100.0 | | |
| G1 Haplotype[†] rs73885319 (A>G) + rs60910145 (T>G) | | | | | | |
| G1 | 43 | 26.7 | 30 | 52.6 | 0.33 [0.17 to 0.62] | 0.0005 |
| Non-G1 | 118 | 73.3 | 27 | 47.4 | | |
| Total | 161 | 100.0 | 57 | 100.0 | | |
| G2 Haplotype rs71785313 (TTATAA>del6) | | | | | | |
| G2 | 59 | 35.3 | 9 | 15.0 | 3.08 [1.45 to 7.06] | 0.0025 |
| Non-G2 | 108 | 64.7 | 51 | 85.0 | | |
| Total | 167 | 100.0 | 60 | 100.0 | | |

[*]Two-tailed Fisher's exact test with mid-P method using a dominant genetic model (carriage of 1 or 2 copies of the designated *APOL1* haplotype),

[†]Individuals with a partial G1 haplotype were excluded from the analysis. *T.b.g: T. b. gambiense*, OR: odds ratio, CI: confidence interval. Raw data for **Table 3** can be found in **Table 3—source data 1**. An association analysis of the two individual component SNPs of the G1 haplotype can be found in **Table 3—source data 2**.

Source data 1. *APOL1* genotype data for *T.b. gambiense* clinical stage trypanosomiasis patients and latent carriers [*]Individuals excluded from the *APOL1* G1 association analysis. *T.b.g: T. b. gambiense*, G0: genotype compatible with the non-risk G0 allele for both rs73885319 and rs60910145, G1: genotype compatible with the G1 CKD risk allele for both rs73885319 and rs60910145, G1$^M$: genotype compatible with the G1 CKD risk allele for rs60910145 and the non-risk G0 allele for rs73885319, G1$^G$: genotype compatible with the G1 CKD risk allele for rs73885319 and the non-risk G0 allele for rs60910145, G2: genotype compatible with the G2 CKD risk allele for rs71785313.

Source data 2. Association between individual *APOL1* G1 kidney disease risk variants and *T.b. gambiense* infection outcome Two-tailed Fisher's exact test with mid-P method using a dominant genetic model (carriage of 1 or 2 copies of the designated *APOL1* SNP). CKD: chronic kidney disease, *T.b.g: T. b. gambiense*, OR: odds ratio, CI: confidence interval. Raw data for **Source data 2** can be found in **Source data 1**

defined by the potential to express the serum-resistance-associated (SRA) protein (*Xong et al., 1998*; *De Greef and Hamers, 1994*) which binds to ancestral APOL1 (G0), inhibiting its formation of lethal pores in trypanosome membranes (*Vanhamme et al., 2003*; *Pérez-Morga et al., 2005*; *Molina-Portela et al., 2005*; *Thomson and Finkelstein, 2015*; *Vanwalleghem et al., 2015*). The two-amino acid deletion that characterises the G2 haplotype (rs71785313, [p.N388_Y389del]), is situated within a C-terminal region of APOL1 demonstrated to be essential for SRA binding (*Lecordier et al., 2009*) (residues 370–392; *Figure 1*). Studies indicate that G2 shifts the position of a critical lysine residue within the binding region that virtually abolishes the interaction with SRA (*Genovese et al., 2010*; *Thomson et al., 2014*). This implicates evasion of the SRA virulence protein as the probable mechanism by which G2 restores APOL1 lytic function and protects the host against *T.b. rhodesiense* infection. The results of this case-control study add substantial support to the proposed heterozygous advantage model of dominant protection against *T.b. rhodesiense* infection for this recessive kidney disease risk variant.

For the G1 variant, no protective association against *T.b. rhodesiense* infection was detected. This finding is somewhat at odds with the reported moderate in vitro *T.b. rhodesiense* lytic activity for G1 donor plasma and recombinant protein (*Genovese et al., 2010*), and delayed parasitemia in an *APOL1* G1 mouse model (*Thomson et al., 2014*). Notably however, the trypanolytic effect for G1

**Table 4.** Conditional association between kidney disease risk variants and *T.b. gambiense* infection outcome excluding compound heterozygotes

| APOL1 haplotype | Dominant model – infection outcome | | | | | |
| | T.b.g Disease | | T.b.g Carriage | | Association analysis[*] T.b.g Disease/Carriage | |
| | Number | % | Number | % | OR [95% CI] | P |
|---|---|---|---|---|---|---|
| G1 Haplotype[†,‡] rs73885319 (A>G) + rs60910145 (T>G) | | | | | | |
| G1 | 36 | 23.4 | 25 | 48.1 | 0.33 [0.17 to 0.64] | 0.0012 |
| Non-G1 | 118 | 76.6 | 27 | 51.9 | | |
| Total | 154 | 100.0 | 52 | 100.0 | | |
| G2 Haplotype[‡] rs71785313 (TTATAA>del6) | | | | | | |
| G2 | 50 | 31.6 | 4 | 7.3 | 5.87 [2.16 to 20.01] | 0.0001 |
| Non-G2 | 108 | 68.4 | 51 | 92.7 | | |
| Total | 158 | 100.0 | 55 | 100.0 | | |

[*]Two-tailed Fisher's exact test with mid-P method using a dominant model (carriage of 1 or 2 copies of the designated *APOL1* haplotype),

[†]Individuals with a partial G1 haplotype were excluded from the analysis.

[‡]Individuals with a compound heterozygote genotype (G1/G2) were excluded from the analysis. *T.b.g: T. b. gambiense*, OR: odds ratio, CI: confidence interval. Raw data for **Table 4** can be found in **Table 4—source data 1**. An association analysis of the two individual component SNPs of the G1 haplotype can be found in **Table 4—source data 2**.

Source data 1. *APOL1* genotype data for *T. b. gambiense* clinical stage trypanosomiasis patients and latent carriers, excluding compound heterozygotes [*]Individuals excluded from the *APOL1* G1 association analysis. *T.b.g: T. b. gambiense*, G0: genotype compatible with the non-risk G0 allele for both rs73885319 and rs60910145, G1: genotype compatible with the G1 CKD risk allele for both rs73885319 and rs60910145, G1[M]: genotype compatible with the G1 CKD risk allele for rs60910145 and the non-risk G0 allele for rs73885319, G1[G]: genotype compatible with the G1 CKD risk allele for rs73885319 and the non-risk G0 allele for rs60910145, G2: genotype compatible with the G2 CKD risk allele for rs71785313.

Source data 2. Association between individual *APOL1* G1 kidney disease risk variants and *T. b. gambiense* infection outcome, excluding compound heterozygotes Two-tailed Fisher's exact test with mid-P method using a dominant genetic model (carriage of 1 or 2 copies of the designated *APOL1* SNP). Individuals with a compound heterozygote genotype (G1/G2) were excluded from the analysis. CKD: chronic kidney disease, *T.b.g: T. b. gambiense*, OR: odds ratio, CI: confidence interval. Raw data for **Source data 2** can be found in **Source data 1**

in both studies was significantly inferior to the G2 variant by up to several orders of magnitude. The G1 haplotype is composed of two closely positioned missense mutations (rs73885319; [p.S342G] and rs60910145; [p.I384M], *Figure 1*), of which the latter is also located in the crucial SRA-binding region. However, the rs60910145 point mutation results in an isoleucine to methionine substitution that only slightly weakens SRA-APOL1 interaction (*Genovese et al., 2010*) and this substitution alone did not extend survival in a mouse model (*Thomson et al., 2014*). Together these data suggest that the G1 variant is not able to confer substantial protection from *T.b. rhodesiense* infection. However, our data do not preclude an effect of G1 on the time course or severity of *T.b. rhodesiense* disease. Additionally, for both the G1 variant and the small number of G2-possessing individuals that were infected with *T.b. rhodesiense* it is possible that dosage effects or inactivating mutations may be present which have abrogated the trypanolytic ability of these *APOL1* variants. No stop codons or mutual non-synonymous variants were observed within the exomes of these individuals during sequence verification of the APOL1 G1 and G2 genotypes (*Supplementary file 1*), but this possibility cannot be definitively excluded.

In contrast to *T.b. rhodesiense*, *T.b. gambiense* primarily infects humans, and is the pathogen responsible for the majority of human disease (*Simarro et al., 2010*) and significant and widespread epidemics of a slower progressing form of sleeping sickness in Central and West Africa. No lytic ability for either *APOL1* variant was reported against *T.b. gambiense* tested by in vitro assays with donor plasma or recombinant protein (*Genovese et al., 2010*). Consistent with this observation, in

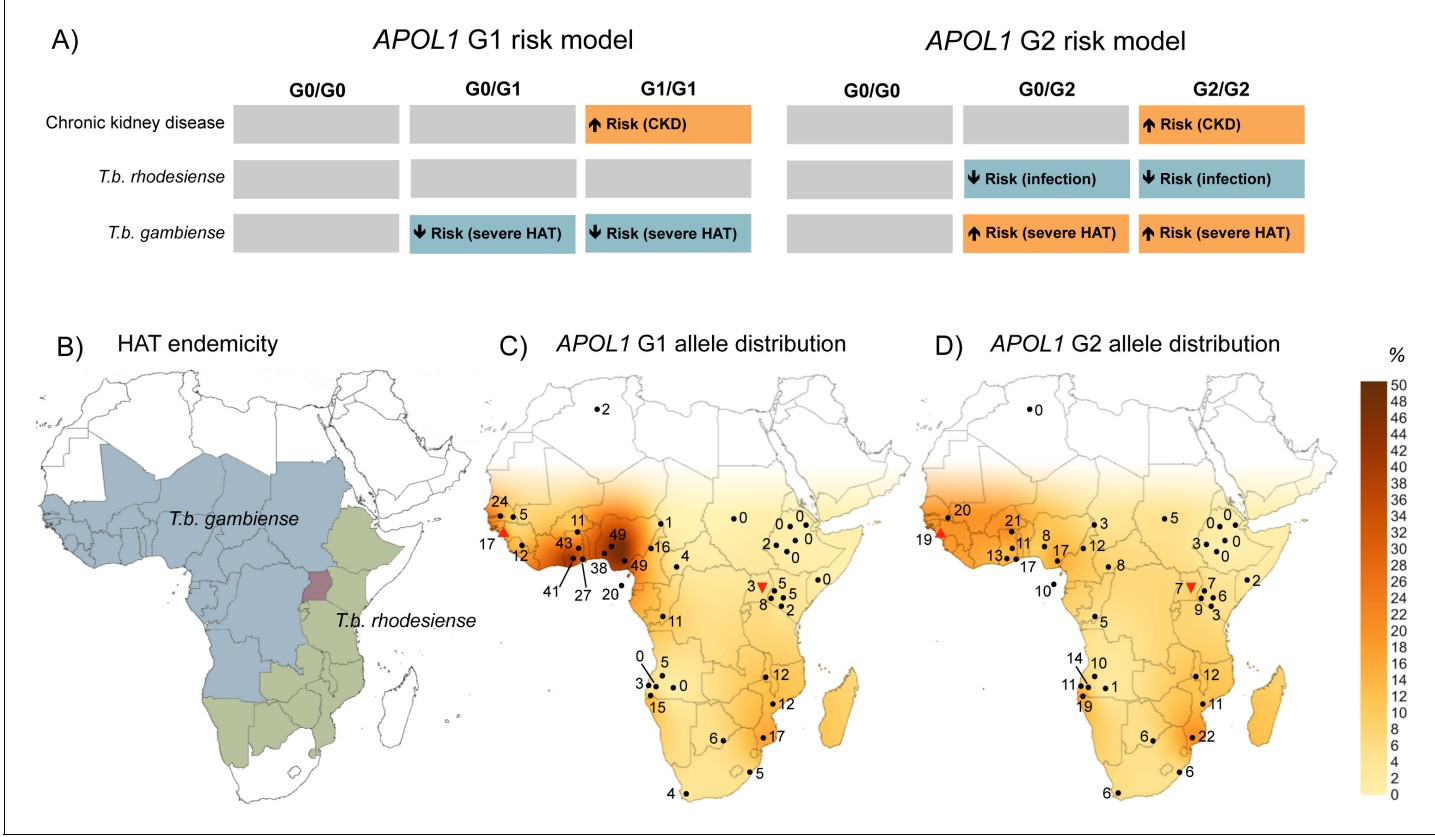

**Figure 2.** The geographical distribution of human African trypanosomiasis and *APOL1* G1 and G2 allele frequencies across sub-Saharan Africa. (**A**) The risk model for chronic kidney disease, *T.b. rhodesiense* infection, and *T.b. gambiense* disease outcome are summarized for the ancestral G0 *APOL1* variant and heterozygous and homozygous carriers of the G1 and G2 variants. The direction of the risk association is indicated by arrow orientation and box colour: orange (increased risk), blue (reduced risk) and grey (no association). (**B**) WHO defines 36 countries as endemic for HAT, caused by *T.b. gambiense* in West Africa (blue) and *T.b. rhodesiense* in East Africa (green). Uganda is the only country endemic for both subspecies, although their distribution does not currently overlap (red). (**C**) Spatial frequency map of the *APOL1* G1 variant. (**D**) Spatial frequency map of the *APOL1* G2 variant. Spatial frequency maps were generated from merged published genotype data available for 40 populations (5287 individuals) in 21 countries (*Figure 2—source data 1*). Colour gradients illustrating predicted allele frequencies across Africa were extrapolated from available data using the Kriging algorithm in Surfer software version 8. The approximate locations of data points are indicated by filled black circles, a filled red triangle (Guinea study), or an inverted filled red triangle (Uganda study) next to the relative allele frequency, in percentage.

The following source data is available for figure 2:

**Source data 1.** Frequency of *APOL1* G1 and G2 variants in African populations.

the Guinea focus neither variant demonstrated a resistance association with *T.b. gambiense* infection. Instead, as summarized in *Figure 2A*, contrasting associations were observed with infection outcome. *APOL1* G1 was associated with a predisposition to latent asymptomatic carriage, while individuals possessing the G2 variant were more likely to progress to clinical disease. The association between *APOL1* variants and infection outcome for *T.b. gambiense* implicates this molecule as a critical modulating factor in disease control. APOL1 is a high-density lipoprotein-associated serum protein, the expression of which is up-regulated by pro-inflammatory stimuli including IFN-γ (*Sana et al., 2005*) and TNF-α (*Monajemi et al., 2002*). In accordance with this, *APOL1* expression is demonstrably increased during *T.b. gambiense* infection (*Ilboudo et al., 2012*). However, no association has been observed between *APOL1* expression levels and blood parasite density or clinical outcome of *T.b. gambiense* infection (*Ilboudo et al., 2012*). Instead, the results of our study indicate that particular *APOL1* variants, rather than modulation of global APOL1 protein level, contribute to differential susceptibility to disease. How these variants influence *T.b. gambiense* is less perceptible than for *T.b. rhodesiense*. The mechanism of *T.b. gambiense* APOL1 resistance does not involve

SRA, but three independent contributing components have been implicated: a sub-species-specific protein, TgsGP (*Capewell et al., 2013*; *Uzureau et al., 2013*), which alters trypanosome membrane rigidity (*Uzureau et al., 2013*); reduced uptake of APOL1 (*DeJesus et al., 2013* ; *Kieft et al., 2010*); and proposed faster degradation of APOL1 within the endocytic system of the parasite (*Uzureau et al., 2013*; *Alsford et al., 2014*). It is possible that alterations to the APOL1 molecule conferred by G1 and G2 polymorphisms affects one or more of these processes with opposing downstream consequences. Furthermore, the strengthened risk association of G2 with clinical disease when individuals who possess both haplotypes (G1/G2 compound heterozygotes) were excluded indicates a potential dominance for the protective G1 haplotype that might be able to mitigate the disease progressive effects of the G2 variant.

Despite its critical function in human innate resistance to most trypanosomes, the role of APOL1 in *T.b. gambiense* disease progression appears complex. Contrasting inflammatory cytokine profiles reported between individuals with clinical stage disease and latent carriers (*Ilboudo et al., 2014*) suggests that an intricate multi-gene interplay between host immune factors, APOL1, and the parasite ultimately determines disease outcome for this subspecies.

For the G1 variant, the relationship with *T.b. gambiense* appears more akin to the well- established association between *Plasmodium* and the sickle haemoglobin S (*HbS*) polymorphism (*Allison, 1954*). In this classic example of heterozygous advantage, the heterozygous *HbS* genotype does not protect from *Plasmodium* infection per se but reduces the risk of severe malaria once infected (*Allison, 1954*; *Taylor et al., 2012*). This advantage has selected and maintained prevalence of the *HbS* polymorphism in malaria-endemic sub-Saharan Africa, despite the high penetrance of life-threatening sickle cell disease in homozygotes (*Allison, 1954*; *Piel et al., 2010*). In *T.b. gambiense*, possession of a G1 allele is associated with the capacity to sustain the asymptomatic latent period of what is normally a fatal disease. This moderation of disease severity could plausibly confer greater survival and reproductive opportunities for individuals possessing the G1 variant than for their G0- or G2-carrying counterparts, who typically progress more rapidly to severe disease (G2 > G0 > G1). Such a selection advantage may explain the high allele frequency of G1 recorded in *T.b. gambiense*-endemic West Africa (up to 49% (*Thomson et al., 2014*; *Abecasis et al., 2012*); *Figure 2C*), which in some populations exceeds even the maximum global *HbS* alleles frequencies (*Piel et al., 2010*). This is consistent with a strong positive selective force on G1 (*Genovese et al., 2010*), and conceivably, a less powerful opposing deleterious pressure from kidney disease in homozygotes, which is typically of late onset, and incompletely penetrant (*Kruzel-Davila et al., 2016*).

Population genetics studies of *T. brucei* indicates that both human-infective sub-species likely arose independently and relatively recently from the animal pathogen *T.b. brucei* (*Tait et al., 1985*; *Gibson et al., 2002*; *Balmer et al., 2011*; *Weir et al., 2016*). Molecular clock analysis dates the emergence of *T.b. gambiense* as a human pathogen in West Africa from a single progenitor approximately 1,000–10,000 years ago (*Weir et al., 2016*). During this time a pivotal lifestyle transition was occurring with the development of agriculture and larger, more densely populated permanent settlements that provided favorable conditions for the emergence of many animal-derived human pathogens (*Harper and Armelagos, 2010*; *Wolfe et al., 2007*). This also coincides with the timeline for a robust selective sweep on G1 detected in the Nigerian Yoruba population (*Genovese et al., 2010*), at the geographical hotspot for this allele (*Figure 2C*) in West Africa. A plausible scenario is that within the last 10,000 years an animal-infective *T.b. brucei* predecessor of *T.b. gambiense* evolved the essential human serum resistance gene *TgsGP* (*Capewell et al., 2013*), facilitating its transmission to humans in the ancestral Bantu population of the Nigeria-Cameroon region. Over time, as *T.b. gambiense* has undergone progressive adaptation into a predominantly human pathogen (*Wolfe et al., 2007*), selection for the human *APOL1* G1 variant may have occurred in turn, which was able to mitigate the lethal progression of disease and promote long-term asymptomatic carriage. The *T.b. gambiense*-protective *APOL1* G1 haplotype could then have spread with human migration and introgression into other sub-Saharan populations during the Bantu expansions (*Tishkoff et al., 2009*), or along commercial routes within the last 4000 years, to reach its current distribution across sub-Saharan Africa (*Figure 2C*).

For *APOL1* G2, the increased risk of clinical *T.b. gambiense* disease contrasts with the strong protective association observed for this variant against *T.b. rhodesiense*. Puzzlingly, as for G1, some of the highest frequencies of G2 are also found in *T.b. gambiense*-endemic West Africa (*Figure 2D*), raising speculation about the evolutionary history of these two variants. Studies of genetic diversity

at the *APOL1* locus are consistent with a older (2,000–7,000 years), less intensively selected allele (*Genovese et al., 2010*; *Pinto et al., 2016*) for G2, and a more recent, rapid sweep for the G1 allele in West Africa (*Genovese et al., 2010*; *Pinto et al., 2016*; *Limou et al., 2015*). One possible interpretation of the available data is that *T.b. rhodesiense* preceded *T.b. gambiense* in West Africa and was responsible for driving positive selection of the G2 variant in the Nigeria-Cameroon region. Rising frequencies of this protective variant (or other unrelated epidemiological factors) could have then forced an eastward shift in *T.b. rhodesiense* endemicity to an approximation of its current distribution in East and Southern Africa. Subsequently, when *T.b. gambiense* emerged in West Africa, the relative fitness of the *APOL1* G2 allele in the exposed population would have been diminished, providing an opportunity for the robust selective sweep of an alternate *APOL1* variant, G1, which was able to reduce the disease severity of this new pathogen.

While this is an attractive theory, there is little epidemiological support for a shift in the endemicity of *T.b. rhodesiense*, which has only been detected in East Africa and has no recorded history in West Africa (*Gibson et al., 2002*; *Radwanska et al., 2002*; *Balyeidhusa et al., 2012*; *Picozzi et al., 2005*). Moreover, isolates of *T.b. rhodesiense* from across East Africa show a strong genetic relationship with sympatric *T.b. brucei* strains, compatible with a predominantly East African origin (*Balmer et al., 2011*; *Godfrey et al., 1990*; *MacLeod et al., 2001*). An alternative model is that selection in favour of the G2 variant may have originated from a different source in West Africa, and it is only more recently, as the G2 variant spread eastwards with the Bantu expansion (*Tishkoff et al., 2009*), that it has fortuitously proved advantageous against *T.b. rhodesiense*. Indeed, beyond its proven capacity for trypanolysis, APOL1 was shown to limit *Leishmania major* infections in mice (*Samanovic et al., 2009*) and suppress HIV-replication in macrophages (*Taylor et al., 2014*), hinting at a much broader role for APOL1 in innate immunity to infectious disease.

The association between *APOL1* chronic kidney disease risk variants and human African trypanosomiasis reveals a more complex picture of selection and human evolution than was originally hypothesized. Despite their close genetic proximity *APOL1* G1 and G2 polymorphisms confer very different, and even opposing, dominant associations with human African trypanosomiasis susceptibility, yet appear convergent in their deleterious recessive contribution to kidney pathology. While the origins of the G2 allele remain speculative, a model of dominant protection against *T.b. rhodesiense* infection is supported. For G1, the strong West African allele distribution bias and evidence for recent, rapid, positive selection, suggest an alternative evolutionary ancestry for this allele, which we propose involves protection from the lethal consequences of the *T.b. gambiense* parasite.

## Materials and methods

### Ethics statement

Participants were identified through healthcare providers, community engagement and active field surveillance in association with the national control programmes. Written informed consent for sample collection, analysis and publication of anonymised data was obtained from all participants by trained local healthcare workers. Subjects or their legal guardian gave consent as a signature or a thumbprint after receiving standardised information in English, French or their local language, as preferred, and were free to withdraw from the study at any time. Efforts were made to ensure the engagement of all local stake holders and approval was obtained from local leaders in each study area where appropriate. Ethical approvals for the study were obtained from within the TrypanoGEN Project following H3Africa Consortium guidelines for informed consent (*H3Africa Consortium, 2013*), from Comité Consultatif de Déontologie et d'Ethique (CCDE) at the Institut de recherche pour le développement (IRD; 10/06/2013) for the Guinea study, and from the Uganda National Council for Science and Technology (UNCST; 21/03/2013) for the Uganda study. Research procedures were also approved by the University of Glasgow MVLS Ethics Committee for Non-Clinical Research Involving Human Subjects (Reference no. 200120043).

## Sample collection

### Uganda

A *T.b. rhodesiense* cohort of 184 blood samples was collected from patients presenting to local hospitals during an epidemic in the neighbouring districts of Soroti and Kaberamaido in Central Eastern Uganda, along with 180 controls, between 2002 and 2012. The majority of the population is from the Kumam ethnic group. In all cases, *T.b. rhodesiense* infection was confirmed by microscopic detection of trypanosomes in wet blood films, Giemsa stained thick blood films or in the buffy coat fraction after microhaematocrit centrifugation. Blood was collected by venepuncture from consenting participants, and preserved as blood spots on FTA filter cards (Whatman, NJ, USA) with air-drying. For PCR amplification, discs of 2 mm diameter were cut from each blood spot using a Harris Micro-punch (Whatman) and prepared according to the instructions provided by the manufacturer.

### Guinea

Samples were collected from a group of three closely positioned active *T.b. gambiense* HAT foci (Dubreka, Boffa and Forecariah) located in the mangrove area of coastal Guinea (*Camara et al., 2005*). The majority of the population is from the Soussou ethnic group. All subjects included in this study were identified during medical surveys performed between 2007 and 2011 by the National Control Program according to standard procedures described elsewhere (*Ilboudo et al., 2011*). For each study participant, 100 µL of plasma and 500 µL of buffy coat were taken. All samples were frozen in the field at −20°C. The highly specific *T. b. gambiense* immune trypanolysis (TL) test was performed on plasma samples as previously described (*Jamonneau et al., 2010*). We included 331 individuals in three phenotypic categories: (i) HAT patients (n = 167): card agglutination test for trypanosomiasis (CATT) positive and trypanosomes detected by the mini Anion Exchange Centrifugation technique (mAECT) followed by microscopy and / or examination of cervical lymph node aspirates by microscopy when adenopathies were present, (ii) Latent carriers (n = 60) CATT plasma titre 1/8 or higher; TL positive, no trypanosomes detected by mean of mAECT and / or examination of cervical lymph node aspirates during a two-year follow-up; (iii) Uninfected endemic controls (n = 104), CATT negative, TL negative, mAECT negative. DNA was extracted from blood collected in the field with the DNeasy Tissue kit (Qiagen, Germany) according to the instructions provided by the manufacturer.

## *APOL1* genotyping

### Uganda

The *APOL1* genotype of each individual at the G1 and G2 loci was determined using PCR–restriction fragment length polymorphism (RFLP) analysis. G1 comprises two non-synonymous substitutions, rs73885319 (c.1024A>G [p.Ser342Gly]) and rs60910145 (c.1152T>G [p.Ile384Met]) in near-perfect linkage disequilibrium. The second variant, G2, is found on an alternative haplotype and represents a two amino acid in-frame deletion (c.1164_1169del [p.Asn388_Tyr389del]). Prepared FTA card discs were used as template in a PCR amplifying a 458 bp product containing the three known variant sites (primers: *APOL1* F1, 5'- AGACGAGCCAGAGCCAATCTTC-3' and *APOL1*_R2, 5'- CACCA TTGCACTCCAACTTGGC −3'). PCR reactions were prepared in a volume of 25 µL using conditions previously described (*Cooper et al., 2008*) with a final primer concentration of 10 µM and 1 unit of Taq polymerase (ThermoFisher Scientific, MA, USA). Amplification was performed using 35 cycles of 95°C for 50 s, 67°C for 50 s, and 70°C for 1 min. Following PCR amplification, an independent RFLP assay was performed for each of the three polymorphisms. For SNP rs73885319 (G1[S342G]), the A>G substitution results in the loss of a HindIII site, for SNP rs60910145 (G1[I384M]) the T>G substitution creates an NspI site, and for rs71785313, the G2 6 bp deletion results in the loss of an MluCI site. For each reaction, 2 µL of PCR product was digested with 10 units of enzyme and the products separated by electrophoresis on a 2% agarose gel. Data for each SNP (rs73885319, rs60910145, and rs71785313) were combined to generate the *APOL1* genotype for each individual. All individuals that were identified as containing G1 or G2 polymorphisms by RFLP, along with a similar number of randomly selected G0 homozygous individuals were verified by PCR amplification and Sanger sequencing (MWG-Biotech AG, Germany) of *APOL1* protein coding exons 3–7. Sequences were evaluated using CLC genomics software (RRID:SCR_011853) for genetic variants relative to NCBI

Genome reference build 38.7 (RRID:SCR_006553; *Supplementary file 1*). Details of the PCR and sequencing primers are provided in *Supplementary file 2*.

## Guinea

The G1$^{S342G}$ (rs73885319) polymorphism was detected by PCR-RFLP. PCR was carried out in a total volume of 30 µL containing 100 ng of DNA, 10 mM of dNTP, 10 µM of each primer (*APOL1* 319_1F: 5'-CAGCATCCTGGAAATGAGC-3'; *APOL1* 319_1R: 5'-GCCCTGTGGTCACAGTTCTT-3') and 1 unit of Taq polymerase (MP Biomedical, CA, USA). The PCR conditions were: 95°C for 5 min followed by 35 cycles at 95°C for 30 s, 59°C for 30 s, 72°C for 45 s and one cycle of extension, 72°C for 5 min. The PCR products were then digested by fast digest HindIII (ThermoFisher Scientific) and digested fragments separated on 2% agarose gel electrophoresis. The G1$^{I384M}$ (rs60910145) polymorphism was genotyped by the Genome and Transcriptome Platform of Bordeaux using the Sequenom MassARRay iplex method. Genotypes were identified with the MassARRAy Typer 4.0 Analyzer software. The G2 (rs71785313) indel was detected on a Li-Cor sequencer. The PCR primers were the same as for G1$^{S342G}$ (rs73885319) but the forward primer had a M13 tail (M13 = 5'-CACGACGTTGTAAAAC-GAC-3'). PCRs were carried out in a total volume of 20 µL containing 25 ng of DNA, 10 mM of dNTP, 10 µM of each primer, 10 mM of dye (M13IR700) and 1 unit of Taq polymerase (MP Biomedical). The PCR conditions were: 95°C for 5 min followed by 35 cycles at 95°C for 30 s, 59°C for 30 s, 72°C for 45 s and one cycle of extension, 72°C for 5 min. The PCR products were then visualized on the Li-Core sequencer for the G2 indel detection. Data for each SNP (rs73885319, rs60910145, and rs71785313) were combined to generate the *APOL1* genotype for each individual.

## Statistical analyses

Statistical analyses of association between *APOL1* genotypes and human African trypanosomiasis in this case-control study were performed by contingency table analyses using Fisher's exact test with mid-P method. Statistical tests were computed using Open-epi. Calculation of the minimum detectable odds ratios was performed for the study sample size in Uganda (<0.144, >2.662 [G1], < 0.350, >2.116 [G2]) and Guinea (<0.448, >2.008 [G1], <0.471, >1.971[G2]) using Sampsize software with the parameters of 80% power, 5% alpha risk and a two-sided test.

## Spatial frequency map of G1 and G2 allele frequency and human African trypanosomiasis distribution

To visualize the geographical distribution of *APOL1* G1 and G2 polymorphisms in sub-Saharan Africa, a contour map was generated by collating data from this study with previously published datasets to produce a cohort of 5287 individuals from across 40 African populations. Published datasets with a low sample size (n ≤ 19) were excluded. G1$^{S342G}$ (rs73885319) was used as a proxy for G1, where G1 frequency data were unavailable (rs73885319 and rs60910145 are in almost complete positive linkage disequilibrium) (*Genovese et al., 2010*; *Kopp et al., 2011*). The contour map was drawn using Surfer 8.0 (Golden Software Inc., Golden, Colorado) applying the Kriging algorithm for data interpolation. Interpolation may be inaccurate where there are few data points. G1 and G2 allele frequencies were analysed for an association with the geographical coordinates (absolute latitude and longitude) using Pearson's correlation test (GraphPad Prism version 6.0, RRID:SCR_002798). The map of *T.b. rhodesiense* and *T.b. gambiense* endemicity was drawn from the Human African trypanosomiasis endemicity classification of the Global Health Observatory data repository (*World Health Organization, 2015*)

## Acknowledgements

This project is supported by a Wellcome Trust Senior Research Fellowship awarded to AML (095201/Z/10/Z), the Institut de Recherche pour le Développement (IRD), the French Ministry of Foreign Affairs through the FSP/REFS project, the World Health Organization (WHO), and Wellcome Trust funding awarded to the TrypanoGEN Consortium, members of H3Africa (099310). The funders had no role in study design, data collection and analysis, or preparation of the manuscript. We thank the staff from the HAT National Control Program of Guinea for their help in sampling. We gratefully acknowledge colleagues at the University of Glasgow for critical reading of the manuscript.

# Additional information

## Funding

| Funder | Grant reference number | Author |
|---|---|---|
| Wellcome | 095201/Z/10/Z | Anneli Cooper<br>William Weir<br>Paul Capewell<br>Annette MacLeod |
| Ministère des Affaires Étrangères | | Hamidou Ilboudo<br>Sophie Ravel |
| World Health Organization | | Mamadou Camara<br>Oumou Camara |
| Wellcome | 099310 | Hamidou Ilboudo<br>V Pius Alibu<br>John Enyaru<br>Harry Noyes<br>Mamadou Camara<br>Vincent Jamonneau<br>Enock Matovu<br>Bruno Bucheton<br>Annette MacLeod |

The funders had no role in study design, data collection and interpretation, or the decision to submit the work for publication.

## Author contributions

AC, Formal analysis, Investigation, Methodology, Writing—original draft, Writing—review and editing; HI, Formal analysis, Investigation, Methodology, Writing—review and editing; VPA, MC, Resources, Writing—review and editing; SR, JM, Investigation, Writing—review and editing; JE, Supervision, Writing—review and editing; WW, Formal analysis, Writing—review and editing; HN, PC, Writing—original draft, Writing—review and editing; VJ, Conceptualization, Investigation, Writing—review and editing; OC, Project administration, Writing—review and editing; EM, Conceptualization, Resources, Funding acquisition, Writing—review and editing; BB, Conceptualization, Resources, Funding acquisition, Investigation, Writing—review and editing; AM, Conceptualization, Funding acquisition, Investigation, Writing—original draft, Writing—review and editing

## Author ORCIDs

Anneli Cooper, http://orcid.org/0000-0002-1159-142X

Annette MacLeod, http://orcid.org/0000-0002-0150-5049

## Ethics

Human subjects: Participants were identified through healthcare providers, community engagement and active field surveillance in association with the national control programmes. Written informed consent for sample collection, analysis and publication of anonymised data was obtained from all participants by trained local healthcare workers. Subjects or their legal guardian gave consent as a signature or a thumbprint after receiving standardised information in English, French or their local language, as preferred, and were free to withdraw from the study at any time. Efforts were made to ensure the engagement of all local stake holders and approval was obtained from local leaders in each study area where appropriate. Ethical approvals for the study were obtained from within the TrypanoGEN Project following H3Africa Consortium guidelines for informed consent, from Comité Consultatif de Déontologie et d'Ethique (CCDE) at the Institut de recherche pour le développement (IRD; 10/06/2013) for the Guinea study, and from the Uganda National Council for Science and Technology (UNCST; 21/03/2013) for the Uganda study. Research procedures were also approved by the University of Glasgow MVLS Ethics Committee for Non-Clinical Research Involving Human Subjects (Reference no. 200120043).

## Additional files

**Supplementary files**

• Supplementary file 1. Sequence variants identified in protein-coding exons of the *APOL1* gene in *T.b.r* cases and controls. Sequence variants relative to human genome reference (NCBI Genome browser build 38, RRID:SCR_006553) were identified in exon sequences of individuals identified with a G1 or G2 genotype, and a representative number of G0 individuals. S: synonymous SNP, NS: non-synonymous SNP, *T.b.r: T.b. rhodesiense*. The chromosome 22 position is indicated based on NCBI Genome reference build 38.7 and reference SNP ID if present, is indicated as described by dbSNP (RRID:SCR_002338). The SNP genotype of each individual is described by the appropriate IUPAC nucleotide code. Rs71785313 (G2) genotypes are described by the presence (TTATAA) or absence (del6) of the six base-pair sequence that is deleted in the G2 variant. Due to limited sample availability, sequence data could not be obtained for two G0/G2 individuals (LIL039 and CT059).

• Supplementary file 2. Primer information for PCR amplification and sequencing of *APOL1* protein-coding exons

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
