## [Decision Letter]

Thank you for submitting your article "*APOL1* renal risk variants have contrasting resistance and susceptibility associations with African trypanosomiasis" for consideration by *eLife*. Your article has been favorably evaluated by Mark McCarthy (Senior Editor) and three reviewers, one of whom, Sarah Tishkoff (Reviewer #1), is a member of our Board of Reviewing Editors.

The reviewers have discussed the reviews with one another and the Reviewing Editor has drafted this decision to help you prepare a revised submission.

Summary:

The authors have conducted a case control study of symptomatic/asymptomatic individuals infected by *T.b. rhodesiense* or by *T.b. gambiense* and association with G1 and G2 variants at *APOL1* that have been implicated in increased risk for kidney disease. This study is of considerable interest and impact because the distribution of these variants in Africa, and their possible adaptive significance, has remained an enigma. They are most common in a subset of West African populations and were previously thought to play a role in resistance only to *T.b. rhodesiense* which is common only in East Africa. In this study, they have demonstrated a protective association for G2 against *T.b. rhodesiense* infection. They also observe opposing associations with T.b. gambiense disease outcome-G1 is associated with asymptomatic infection whereas G2 is associated with faster progression of trypanosomiasis. Overall, the study represents a unique and important addition and clarification to the burgeoning scientific and medical research literature on *APOL1* in relation to infectious disease and in relation to kidney disease risk

Essential revisions:

1) For all three tables, the summary information (number and percentage of samples) presents the G1 or G2 genotype. It is not clear if this refers to heterozygotes, such that all those listed as G1 are actually genotype G1/G0, and all of those under G2 are G2/G0 genotype, or whether there are homozygotes or compound heterozygotes included. If the latter it is not the case, it would be important to specify this or use more accurate terminology.

2) The authors seem to have incorrectly added up the amount of G1 *T.b. gambiense* carriers in Table 3. In the raw data, there are 31 people that fit this category, not 30. The authors have either excluded one of either the G1M/G1 or G1G/G1 individuals but not the other with no provided reasoning, or they have simply missed one when calculating. As written in the first paragraph of the subsection “*APOL1* variants and resistance/susceptibility to *T.b. rhodesiense*” "partial G1 were excluded". This should be clarified.

3) Correct annotation in Table 2 and Table 3. For example, sample Bo443-6 in Table 2's raw data is listed as a G1G/G0 despite harboring the del6 at the G2 polymorphism locus. There are five instances of these types of miss annotation between Table 2 and Table 3.

4) Check the calculation for the odds ratio for the tables. We have re calculated all of the odds ratios in each of the tables and both the confidence interval and p value of the G2 odds ratio in Table 3 appears incorrect. We calculate p = 0.0043.

5) Address the possibility that inactivating mutations have abrogated the trypanolytic effect of either or both of the variants, specifically in regard to the small number of (6) of T.b.r. infected G2 genotype individuals, or the 9 T.b.r. infected G1 genotype individuals. The cleanest way to address this would be to sequence through at least the exomes, and if not, to at least add a sentence addressing the extent of implausibility of this scenario. This issue is less relevant for the results summarized in Table 2 and Table 3, since opposing robust associations were found.

6) Include a metric for the odds ratio for the G0 genotype in each of the data tables for comparison to the G1 and G2 genotypes. Secondly, there is some interesting information that is not addressed, such as the fact that the *T.b. gambiense* disease/carriage ratio for people with exclusively the G2/G0 heterozygous genotype is actually 5.008 with a significant p value of 0.0032. Why are these conditional odds ratios not discussed?

7) Please explain in the body of the text why the compound heterozygotes (G1/G2 genotypes) are not being excluded from these analyses, but rather those individuals are being counted for both the G1 and G2 data sets. If G1 is protective, while G2 exacerbates progression of *T. b. gambiense*, then including compound HETs makes very little sense and the reasoning behind their inclusion should be explained.

8) Provide more evidence for the hypothesis that *T.b. gambiense* originated in humans from a species jump and tighten up that discussion so that it is less speculative.

9) Make sure that data source files are accessible.

10) Add an in vitro killing graph with recombinant proteins of all three genotypes to against *T. b. gambiense* and *T.b. rhodesiense* that can be included to show the in vitro effect.

---

## [Author Response]

*Essential revisions:*

*1) For all three tables, the summary information (number and percentage of samples) presents the G1 or G2 genotype. It is not clear if this refers to heterozygotes, such that all those listed as G1 are actually genotype G1/G0, and all of those under G2 are G2/G0 genotype, or whether there are homozygotes or compound heterozygotes included. If the latter it is not the case, it would be important to specify this or use more accurate terminology.*

For all tables, analysis was performed using a dominance model, in which we counted every individual carrying at least one copy of the APOL1 haplotype under analysis (G0, G1 or G2). For example, for the G1 variant, this includes homozygotes (G1/G1), heterozygotes (G1/G0) and compound heterozygotes (G1/G2). We have changed the tables so that exact numbers and percentages both with and without each haplotype are given and added a clarification that this association analysis was performed using a dominant model to each table legend.

Where individuals have been excluded from a particular analysis, for example, individuals with only a partial G1 haplotype have been excluded from the G1 association analysis, this has been detailed in the table legends and in the text (subsection “*APOL1* variants and resistance/susceptibility to *T.b. rhodesiense*”, first paragraph).

We have also made the data for the two component SNPs of the G1 haplotype (rs73885319 and rs60910145) available in the source data (source data 2 for each table) so that the association analysis can be accessed for each SNP individually, in addition to the complete G1 haplotype in the main manuscript.

A separate analysis excluding compound heterozygotes (as suggested by the reviewers in a later comment) has also now been included for *T.b. gambiense* infection outcome (Table 4) and more details of this are given in revision point 7.

*2) The authors seem to have incorrectly added up the amount of G1 T.b. gambiense carriers in Table 3. In the raw data, there are 31 people that fit this category, not 30. The authors have either excluded one of either the G1M/G1 or G1G/G1 individuals but not the other with no provided reasoning, or they have simply missed one when calculating. As written in the first paragraph of the subsection “APOL1 variants and resistance/susceptibility to T.b. rhodesiense” "partial G1 were excluded". This should be clarified.*

A small number of individuals were identified to carry a partial G1 haplotype (the kidney disease risk genotype at one of the G1 polymorphism positions but the non-risk genotype at the other). Individuals possessing only a partial G1 genotype and no full G1 alleles (e.g. G1^G^/G0 or G1^M^/G0) have been excluded from the association analysis of the G1 haplotype. There are 30 individuals that possess the complete G1 haplotype in the *T.b. gambiense* carrier group in Table 3. Examining the source data we can see that one individual carrying the G1^M^/G0 genotype (and thus excluded) was incorrectly labelled as G1^M^/G1 in the “designated APOL1 genotype” column of [Supplementary-material SD5-data]. This was an annotation error and did not affect the statistical calculations. We are grateful to the reviewers for spotting this error and it has now been corrected.

*3) Correct annotation in Table 2 and Table 3. For example, sample Bo443-6 in Table 2's raw data is listed as a G1G/G0 despite harboring the del6 at the G2 polymorphism locus. There are five instances of these types of miss annotation between Table 2 and Table 3.*

As noted for revision point 2, these were unfortunate annotation errors that were restricted to the source data tables and did not affect any of the statistical calculations. We are grateful to the reviewers for spotting this and they have now been corrected.

*4) Check the calculation for the odds ratio for the tables. We have re calculated all of the odds ratios in each of the tables and both the confidence interval and p value of the G2 odds ratio in Table 3 appears incorrect. We calculate p = 0.0043.*

We have recalculated the odds ratio for G2 in Table 3 using a Two-tailed Fisher's exact test with mid-P method and obtained the same odds ratio (3.08) and P value (0.0025) as described in the manuscript. This was calculated using openepi (www.openepi.com) using the numbers provided in Table 3. A screenshot of our statistical analysis is presented in Figure 3.

Author response image 1.**DOI:**
http://dx.doi.org/10.7554/eLife.25461.020

*5) Address the possibility that inactivating mutations have abrogated the trypanolytic effect of either or both of the variants, specifically in regard to the small number of (6) of T.b.r. infected G2 genotype individuals, or the 9 T.b.r. infected G1 genotype individuals. The cleanest way to address this would be to sequence through at least the exomes, and if not, to at least add a sentence addressing the extent of implausibility of this scenario. This issue is less relevant for the results summarized in Table 2 and Table 3, since opposing robust associations were found.*

For the *T.b. rhodesiense* study (Table 1), to confirm genotypes the *APOL1* protein-coding exons 3-7 were sequenced from all G1 or G2 positive individuals (with the exception of 2 individuals [CT059 and LIL039] for which the availability of sample DNA was limited) and a representative number (16) of G0 individuals. This has now been clarified in the Materials and methods section (subsection “Uganda”, last paragraph) and the primer sequences provided in [Supplementary-material SD11-data]. No stop codons were identified in any individual, although a small number of non-synonymous variants were detected. The sequence data with all identified genetic variants has now been included in [Supplementary-material SD10-data]. A sentence addressing this issue and the possibility of inactivating mutations or gene dosage has now been included in the text (Discussion, third paragraph).

*6) Include a metric for the odds ratio for the G0 genotype in each of the data tables for comparison to the G1 and G2 genotypes. Secondly, there is some interesting information that is not addressed, such as the fact that the T.b. gambiense disease/carriage ratio for people with exclusively the G2/G0 heterozygous genotype is actually 5.008 with a significant p value of 0.0032. Why are these conditional odds ratios not discussed?*

*7) Please explain in the body of the text why the compound heterozygotes (G1/G2 genotypes) are not being excluded from these analyses, but rather those individuals are being counted for both the G1 and G2 data sets. If G1 is protective, while G2 exacerbates progression of T. b. gambiense, then including compound HETs makes very little sense and the reasoning behind their inclusion should be explained.*

There is no association for either *T.b. rhodesiense* or *T.b. gambiense* with the G0 genotype. This data has now been included in Table 1–Table 3 for comparison with the G1 and G2 association analyses.

The reviewers raise an interesting point regarding the strong conditional association for the G2 variant with *T.b. gambiense* disease progression. The opposing associations of G1 and G2 were previously unpredicted for *T.b. gambiense.* In light of their identification in this study and the reviewers comments we have now added extra analyses (presented in Table 4) to show the association of the G1 and G2 haplotypes with *T.b. gambiense* disease outcome when compound heterozygotes (G1/G2) are excluded. This analysis strengthens the association of G2 with progression of symptomatic *T.b. gambiense* disease from OR 3.08 [95% CI:1.45 to 7.06], *P* = 0.0025 to OR 5.87 [95% CI: 2.16 to 20.01], *P* =0.0001, suggesting a possible dominance for the protective mechanism of the G1 variant in G1/G2 compound heterozygotes. This data and a brief discussion of its implications has now been added to the text (subsection “Disease outcome”, first paragraph; Discussion, fourth paragraph).

*8) Provide more evidence for the hypothesis that T.b. gambiense originated in humans from a species jump and tighten up that discussion so that it is less speculative.*

In the original manuscript we only referred briefly to the evidence supporting the origin of the human-infective subspecies *T.b. rhodesiense* and *T.b. gambiense* from animal-infective *T.b. brucei*. Both of these subspecies possess a different mechanism of human serum resistance and are suggested to have arisen independently and relatively recently from non-human infective *T.b. brucei*. We have now added references that support this hypothesis for both subspecies, based on population genetics and genomics studies (Tait et al.1985, Gibson et al. 2002, Balmer et al. 2011, Weir et al. 2016). We have also referenced a review by Wolfe et al. (2007) published in Nature that discusses the evolutionary origin and intermediate stages of adaptation of many human–infective pathogens from their animal-infective predecessors, including both of the human-infective *T. brucei* subspecies (Discussion, seventh paragraph).

*9) Make sure that data source files are accessible.*

All the source data files for Table 1–Table 4 have currently been formatted as excel (xlsx) files but we would be happy to adapt these to any alternative format, as required.

*10) Add an in vitro killing graph with recombinant proteins of all three genotypes to against T. b. gambiense and T.b. rhodesiense that can be included to show the in vitro effect.*

This data has been previously published by others (Genovese et al. 2010) and has been referenced in this manuscript. We have changed the text to clarify that both serum and recombinant protein have been previously tested in in vitro assays (Discussion, second, third and fourth paragraphs).